# Avian Louse Flies and Their Trypanosomes: New Vectors, New Lineages and Host–Parasite Associations

**DOI:** 10.3390/microorganisms10030584

**Published:** 2022-03-08

**Authors:** Anežka Santolíková, Jana Brzoňová, Ivan Čepička, Milena Svobodová

**Affiliations:** 1Department of Parasitology, Faculty of Science, Charles University, Vinicna 7, CZ, 128 44 Prague, Czech Republic; jana.brzonova@natur.cuni.cz; 2Department of Zoology, Faculty of Science, Charles University, Vinicna 7, CZ, 128 44 Prague, Czech Republic; ivan.cepicka@natur.cuni.cz

**Keywords:** avian parasite, *Trypanosoma*, transmission, Hippoboscidae, *Ornithomya*, *Ornithoica*, *Ornithophila*, host specificity

## Abstract

Louse flies (Hippoboscidae) are permanent ectoparasites of birds and mammals. They have a cosmopolitan distribution with more than 200 described species. The aim of this study was to reveal host–vector–parasite associations between louse flies, birds, and trypanosomes. A total of 567 louse fly specimens belonging to 7 species were collected from birds at several localities in Czechia, including the rare species *Ornithophila metallica* and *Ornithoica turdi*. There was a significant difference in the occurrence of *Ornithomya avicularia* and *Ornithomya fringillina* on bird hosts according to their migratory status, *O. fringillina* being found more frequently on long-distance migrants. Trypanosomes were found in four species, namely, *Ornithomya avicularia*, *O. fringillina*, *O. biloba*, and *Ornithoica turdi*; the later three species are identified in this paper as natural trypanosome vectors for the first time. The prevalence of trypanosomes ranged between 5 and 19%, the highest being in *O. biloba* and the lowest being in *O. fringillina*. Phylogenetic analysis of the SSU rRNA gene revealed that a vast majority of trypanosomes from hippoboscids belong to the avian *T. corvi/culicavium* group B. Four new lineages were revealed in group B, with louse flies being probable vectors for some of these trypanosome lineages. We also confirmed the transcontinental distribution of several trypanosome lineages. Our results show that hippoboscids of several genera are probable vectors of avian trypanosomes.

## 1. Introduction

Louse flies are ectoparasitic insects found on mammals and birds, with both sexes feeding strictly on blood. They belong to the superfamily Hippoboscoidea, along with the medically important tsetse flies (Glossinidae), vectors of African trypanosomes, and bat flies (Nycteribiidae and Streblidae) [1]. The bodies of louse flies show adaptations to the parasitic lifestyle, including dorsoventrally flattened thorax, head and abdomen, claws for better attachment to the host, and modified mouth parts [2]. Louse flies (Hippoboscidae) have a worldwide distribution and contain 213 species [3]; the majority are parasites of birds, while one quarter of them occur on mammals [4]. Eleven avian louse fly species have been found in Czechia, namely: *Ornithoica turdi*, *Ornithomya avicularia*, *O. chloropus*, *O. fringillina*, *O. biloba*, *Ornithophila metallica*, *Olfersia fumipennis*, *Icosta ardeae*, *Pseudolynchia canariensis*, *Crataerina pallida*, and *Stenepteryx hirundinis* [5]. Louse fly species differ in their geographic distribution: *O. fringillina* and *O. chloropus* are typical for Northern Europe with fewer findings from Central and Southern Europe, whereas *O. turdi* and *O. metallica* have been found, rather, in the southern reaches of the northern hemisphere and are typical for Africa [2].

The host specificity of avian louse flies differs among species, ranging from high in *O. biloba* and *S. hirundinis*, which are essentially limited to a single host species, to low in *Ornithomya avicularia* and *O. fringillina,* which have been found on multiple host genera [6,7,8]. Individual louse flies can even switch their host species when birds are in close contact [9].

Because of their blood-feeding behavior, louse flies are vectors of many pathogens (reviewed in [10]), including trypanosomes. The louse fly *O. avicularia* was identified as a vector of avian trypanosomes after being fed on a trypanosome-infected rook (*Corvus frugilegus*), and subsequently transmitting trypanosomes to a canary (*Serinus canaria*) [11]. Although originally designated as *Trypanosoma avium*, the species used in these experiments is now considered to be *T. corvi* [12]. Since then, studies of trypanosomes in avian louse flies have remained scarce [13,14,15]. Trypanosome prevalence in the louse fly *Ornithomya avicularia* reached 5%, with 90% of the isolates belonging to *T. corvi* [15].

Avian trypanosomes are polyphyletic, splitting into three groups, each consisting of multiple lineages [16]. *T. corvi* is closely related to *T. culicavium* [17], and they both belong to group B, as defined by Zídková et al. [16]. However, trypanosome lineages belonging to the *T. avium* group C have been isolated from louse flies, as well [16,18]. Apart from Hippoboscidae, avian trypanosomes are transmitted by Nematocera, namely, mosquitoes [14,17,19], black flies [20,21], biting midges [22,23,24], and phlebotomine sandflies [25,26]. A particular trypanosome lineage can be transmitted by multiple vectors [16,26].

In this study, we aimed to (i) describe the occurrence of different avian louse fly species in relationship to the host species, (ii) assess the influence of host migratory status on the species of flies they host, (iii) molecularly identify obtained trypanosomes, using the SSU rRNA gene sequences, and (iv) uncover the associations between different species of hippoboscid flies and the trypanosomes they host and transmit. The louse fly vectorial capacity towards different trypanosome groups is discussed. We present complex data concerning this neglected group of kinetoplastid vectors.

## 2. Materials and Methods

### 2.1. Louse Fly Collection, Dissection, and Identification 

Louse flies were collected between 2014 and 2019 from adult birds caught in mist nets and from swallow (*Hirundo rustica*) and swift (*Apus apus*) nestlings by members of the team and other registered ringers in several Czech localities, mainly Zeměchy (50.231783, 14.272371) and Choteč (49.999069, 14.280239) in Central Bohemia and Milovický forest (48.821274, 16.693175) in South Moravia. Collected louse flies were placed in zip-lock bags and stored in a cooling box before dissection. The flies that did not survive until dissection were stored in 96% ethanol at −20 °C.

Live louse flies were killed in 96% ethanol, washed twice in 0.9% sterile saline, and dissected on a glass slide under a stereomicroscope. Wings of the louse flies were separated under a stereomicroscope and mounted on glass slides in CMCP-9 mountant (Polysciences, Warrington, PA, USA). Identification was performed using two keys [2,27].

### 2.2. Trypanosome Cultivation, Microscopy, and Statistics

Trypanosome-positive guts were inoculated on rabbit blood agar, and trypanosomes were cultured as described previously [26]. A part of each positive guts sample was stored in ethanol as a backup for barcoding in case of unsuccessful culture. Thriving trypanosome cultures were frozen in 7% dimethylsulfoxide (final concentration) and stored in liquid nitrogen.

For scanning electron microscopy, a trypanosome-positive gut was treated as described in [26].

Bird migratory status was assigned as described in [28], and the data were processed using R software [29].

### 2.3. DNA Isolation, Amplification, and Sequencing

Ethanol from samples was evaporated in a thermoshaker at 39 °C. Whole louse fly body samples were crushed in Eppendorf tubes using sterile micropestles. DNA from individual samples was then extracted using the High Pure Template Preparation Kit (Roche, Basel, Switzerland) according to the manufacturer’s instructions.

For trypanosome detection, the SSU rRNA gene was amplified using nested PCR with primers S-762 (5′-GACTTTTGCTTCCTCTAWTG-3′) and S-763 (5′-CATATGCTTGTTTCAAGGAC-3′) [30] in the first step. The first amplification round consisted of 35 cycles and was performed in the final volume of 11 µL of PCR mix (EmeraldAmp GT PCR Master Mix (TaKaRa, Shiga, Japan). The annealing temperature was 55 °C. For the second run, consisting of 35 amplification cycles, 1 µL of the product of the first amplification round was used as a template in 24 µL of PCR mix with TR-F2 (5′-GARTCTGCGCATGGCTCATTACATCAGA-3′) and TR-R2 (5′-CRCAGTTTGATGAGCTGCGCCT-3′) primers [31]. The annealing temperature was 64 °C.

Positive PCR products were purified by ExoSAP-IT™ (Thermo Scientific, Waltham, MA, USA) and sequenced in the Core Facility of the Faculty of Science, Charles University, using the primer 1000R (5′-ATGCCTTCGCTGTAGTTCGTCT-3′) [32], resulting in an approximately 600bp-long sequences. Further sequencing of chosen strains (K51, PAS441, and PAS433) was performed with primers 1000F (5′-AGACGAACTACAGCGAAGGCAT-3′) [32] and kin577F (GCCAGCACCCGCGGT) [16]; the assembly for each of the 3 sequences was carried out in Geneious 9.1.7 and was approximately 1500bp long. Low-quality ends were trimmed from all sequences in BioEdit 7.0.4.1. [33].

### 2.4. Sequence Analysis

In total, 137 SSU rRNA gene sequences of trypanosomes were used in the phylogenetic analysis, of which 85 were newly determined: 57 were sequences from louse flies, and 18 represented novel genotypes of avian trypanosomes belonging to group B and obtained from avian blood in a parallel study as described in [26].

The sequences were aligned using the MAFFT method [34] on MAFFT 7 online server (https://mafft.cbrc.jp/alignment/server/, accessed on 10 June 2021) with the G-INS-i algorithm and default settings. The final alignment consisted of 1947 characters. The phylogenetic tree was constructed by the maximum likelihood method in RAxML 8.0.0. [35], under GTRGAMMAI model. Statistical support of the topology was assessed by bootstrapping with 1000 pseudoreplicates in RAxML. The tree was then graphically edited in CorelDRAW X8 and Inkscape 2.1.

## 3. Results

### 3.1. Collected Louse Flies 

In total, 567 louse flies belonging to 7 species were caught: *Ornithomya biloba* (306), *O. avicularia* (133), *O. fringillina* (78), *Ornithoica turdi* (14), *Stenepteryx hirundinis* (2), *Ornithophila metallica* (1), and *Crataerina pallida* (33) (Figure 1, Table 1).

### 3.2. Host Specificity of Hippoboscids

In total, 39 avian species belonging to 28 genera were found to host louse flies; of those, 17 species were long-distance migrants, 13 were short-distance migrants, and 9 were residents. *Ornithomya avicularia* was found on 32 species from 23 genera, *O. fringillina* on 19 species from 12 genera, *O. biloba* on 4 species and genera, and *Ornithoica turdi* was found on 4 species and genera (namely, *Coccothraustes coccothraustes*, *Emberiza citrinella*, *Sitta europaea*, *Turdus philomelos*). A single specimen of *Ornithophila metallica* was found on *C. coccothraustes*. *Stenepteryx hirundinis* was found on two species from two genera of Hirundinidae, and *Crataerina pallida* was found on its specific host species, *Apus apus* (Table 1).

We evaluated the differences in host preferences of the two opportunistic louse fly species, *O. avicularia* and *O. fringillina*, according to the migratory status of their bird hosts. The difference between these preferences is significant (Pearson’s Chi-squared test, X = 34.042, df = 2, *p* < 0.001). Although in different ratios, both species occurred on residents and short- and long-distance migrants (Figure 2).

### 3.3. Prevalence of Trypanosomes in Louse Flies

Four of the seven collected louse fly species harbored trypanosomes in their gut, namely, *Ornithomya biloba, O. avicularia*, *O. fringillina*, and *Ornithoica turdi*. Trypanosome infections were mature, localized in the hindguts, and putative metacyclic forms were observed in the vast majority of cases. The prevalence of trypanosomes differed significantly, with the highest prevalence in *O. biloba* (18.7%), followed by *O. turdi* (7.1%), *O. avicularia* (6.6%), and *O. fringillina* (4.6%; Figure 3). The prevalence of trypanosomes significantly differs among the four louse fly species (Fisher’s Exact Test, *p* < 0.001). No trypanosomes were found in 33 specimens of *Crataerina pallida*, which is specific for swifts.

In the vast majority of cases, trypanosomes massively colonized hippoboscid guts (Figure 4).

### 3.4. Phylogenetic Analysis 

Trypanosome lineages obtained from louse flies belonged to groups B (*T. corvi/culicavium*) and C (*T. avium/thomasbancrofti*). The majority (69) of these trypanosome sequences belonged to group B, whereas only two belonged to group C (Figure 5). Some obtained *Trypanosoma* sequences from louse flies belonged to lineages described previously, while 61 of them belonged to new lineages. Newly determined unique sequences are available in GenBank under accession numbers OM509725-OM509732.

The well-known species transmitted by louse flies is *T. corvi* (lineage IV). Three newly obtained *Trypanosoma* sequences from *O. avicularia* belonged to this lineage, together with previously sequenced trypanosomes from *O. avicularia* and birds (buzzard *Buteo buteo*, rook *Corvus frugilegus*, large-billed crow *C. macrorhynchos*, and currawong *Strepera* sp.) from Europe, Asia, and Australia.

The second described *Trypanosoma* species from the B group is *T. culicavium* (lineage V). In addition to the originally identified avian host is the collared flycatcher (*Ficedula albicollis*). We have found this lineage in several new hosts, namely, the Eurasian reed warbler (*Acrocephalus scirpaceus*), swallow (*Hirundo rustica*), chiffchaff (*Phylloscopus collybita*), yellowhammer (*Emberiza citrinella*), and nightingale (*Luscinia megarhynchos*).

Two new trypanosome sequences were isolated from birds clustered with lineage XII; this lineage consists of only trypanosome sequences from birds; vectors of these trypanosomes remain unknown.

Our phylogenetic analysis revealed four new lineages in the group B, which we designate B13 to B16 (Figure 5).

The lineage B13 consisted of isolates from hippoboscids (*O. avicularia* and *O. biloba*) and passerines. In addition to a song thrush (*Turdus philomelos*) isolate and two isolates from collared flycatcher (*Ficedula albicollis*), this lineage contained a sequence from an Australian passerine yellow spotted honeyeater (*Meliphaga notata*). The lineage B13 was previously identified and is as of yet unnamed [16]. Trypanosome sequences from louse flies clustering with this lineage indicate hippoboscids as newly identified vectors of the lineage B13.

The lineage B14 consisted of trypanosome isolates from hippoboscids (all except one from *O. biloba*, one from *O. fringillina* caught on a sparrow (*Passer montanus*)) and trypanosomes of raptors and owls from Thailand. This lineage differed from the lineage I, its closest relative, by four nucleotides in the sequenced region.

B15 and B16 were ancestral lineages at the base of group B and consisted of only trypanosome sequences originating from Passerines; interestingly, B15 contained our new sequences from European passerines blackcap (*S. atricapilla*), great reed warbler *A. arundinaceus*, and wood warbler *P. sibilatrix*, together with American and Australian birds: wood thrush (*Hylocichla mustelina*), yellow-breasted chat (*Icteria virens*), and ashy robin (*Heteromyias albispecularis*). B16 consisted of a single sequence from the reed warbler *A. scirpaceus*. Vectors of these two lineages remain unknown. The two new lineages B15 and B16 differed in 52 and 23 nucleotides, respectively, from lineage I.

A single new trypanosome sequence obtained from the hippoboscid *Ornithomya fringillina* (OF19 [18]) was placed by our phylogenetic analysis to the group C, lineage II (*Trypanosoma thomasbancrofti*).

## 4. Discussion

Although hippoboscids were already identified as vectors of avian trypanosomes by the middle of the 20th century [11], their vectorial role has scarcely been studied since. The finding of trypanosomes in several louse fly species in our study, as well as the transcontinental distribution of group B trypanosomes transmitted by the hippoboscids, indicate a high potential for louse flies as vectors of avian trypanosomes in other continents as well.

### 4.1. Findings of Avian Hippoboscids

While three common species belonging to the genus *Ornithomya* were present in all catching sites, *Ornithophila metallica* and *Ornithoica turdi* occurred only in the southernmost site of Milovický forest. *O. turdi* was reported from South Moravia previously but was considered non-breeding [2,36]. However, the relatively frequent occurrence, and occurrence on resident birds suggest this species completes its life cycle in Czechia. In fact, we recorded many more sightings of specimens that eluded capture due to their high agility and small size.

*O. metallica* has been reported in Czechia only twice in 1956 and 1973 in South Moravia [2,5], and this species is considered a southern element.

### 4.2. Host Specificity of Avian Louse Flies

The host specificity of avian louse flies differed among louse fly genera and species. Some species are considered strictly host specific; *Crataerina pallida* for the swift, *Stenepteryx hirundinis* for the common house martin *Delichon urbicum*, and swallow *Hirundo rustica* [2,6], as supported by our findings. Similarly, *Ornithomya biloba* is host specific toward Hirundinidae, with the vast majority of specimens found on swallows [2,6]. Nevertheless, we also document occasional findings of *O. biloba* on other avian hosts, namely, blue tit (*Cyanistes caeruleus*) and sparrow (*Passer montanus*). The louse fly species *O. avicularia, O. fringillina*, and *Ornithoica turdi* displayed very low host specificity and have been found on hosts belonging to several passerine families; this corresponds to previous studies [2,6].

### 4.3. Migration Status of Louse Fly Hosts

The geographical distribution of *Ornithomya fringillina* is holarctic, the species being found mainly in Northern Europe, with scarce findings in Central and Southern Europe, presumably on migrants [2]. We compared the host species composition of *O. avicularia* and *O. fringillina* according to the bird migratory status. *O. fringillina* was found most frequently on long-distance migrants, whereas *O. avicularia* was found mostly on residents and short-distance migrants. Although the difference was significant, it was not clear-cut; the question persists whether specimens of *O. fringillina* found on resident birds emerged locally or switched hosts from northern migrants [9]. Interestingly, a recent study in Finland did not find any difference between three *Ornithomya* species abundances in relation to host migratory status [7], which might support the hypothesis that *O. fringillina* found in Czechia originated from northern breeding grounds, given that emerging louse flies attach to the first available avian host specimen, which then brings it to the south during its fall migration.

### 4.4. Trypanosomes in Avian Hippoboscids

To date, the only hippoboscid species found to be naturally infected with trypanosomes was *Ornithomya avicularia*. This species was used for the experimental transmission of trypanosomes [11,13], and trypanosomes were readily found in specimens captured on raptor nestlings [15]. Trypanosomes were found in *O. fringillina* experimentally fed on infected birds and were, in one case, transmitted to the same host species by intraperitoneal injection [20]. In this study, we have found *O. avicularia*, *O. biloba*, *O. fringillina*, and *Ornithoica turdi* naturally infected with avian trypanosomes. Trypanosome infections were mature, localized in the hindguts, and putative metacyclic forms were observed. Therefore, we consider these hippoboscid species as competent vectors. The prevalence of trypanosomes in different louse fly species varied form 18.7% in *O. biloba* to 4.6% in *O. fringillina*. The prevalence of 6.6% in *O. avicularia* is only slightly higher than in previous screening of this hippoboscid species caught on raptors [15].

Although the permanent presence of host blood in the louse fly intestine enables trypanosomes to thrive regardless of the vector capacity of the louse fly, the massive infections of hippoboscid guts indicate a true ability of these insects to transmit avian trypanosomes. Apart from being true vectors, hippoboscids can serve as unspecific hosts for group C trypanosomes transmitted by mosquitoes [18] or blackflies [16], but their potential to transmit them remains to be confirmed.

### 4.5. Phylogeny of Avian Trypanosomes and Host–Parasite Associations

Our phylogenetic analysis was unable to resolve the relationships between all particular lineages within the group C, which is in agreement with previous studies [16].

The lineage B14 is frequently found in hippoboscids, namely, in *Ornithomya biloba* from swallows; the only exception out of 53 isolates from louse flies was a single specimen from *O. fringillina* caught on a sparrow. Interestingly, the trypanosome clade found in *O. biloba*, which exhibited the highest prevalence in louse flies, could not be detected in its avian hosts (59 barn swallows sampled, see [18]). In Thailand, trypanosome sequences clustering to the lineage B14 were obtained from blood samples from owls and raptors [36]. Given the strict host specificity of *O. biloba* and its documented geographic distribution, it is likely that the B14 trypanosome lineage is transmitted to raptors and owls by other hippoboscid species in Thailand.

The lineages in the group B of avian trypanosomes indicate specificity toward the louse fly host species *O. biloba* (lineage B14) and *O. avicularia* (lineage B13, IV and I). The trypanosome sequences isolated from *O. fringillina* were scattered among different lineages (B14, I and lineage II from the group C).

A trypanosome from the only positive *Ornithoica turdi* clusters within the lineage I, which originally contained only two trypanosome sequences from *O. avicularia* [16] but now, in addition to louse fly sequences, also contains lineages from passerines and owls [37], suggesting low host specificity toward both avian and hippoboscid hosts.

## 5. Summary

Avian trypanosomes are transmitted by louse flies belonging to several species and genera. *Ornithomya fringillina* is probably brought from the north on migrating birds, while *Ornithoica turdi* breeds in Central Europe. Avian trypanosomes transmitted by louse flies have a transcontinental distribution.

## Figures and Tables

**Figure 1 microorganisms-10-00584-f001:**
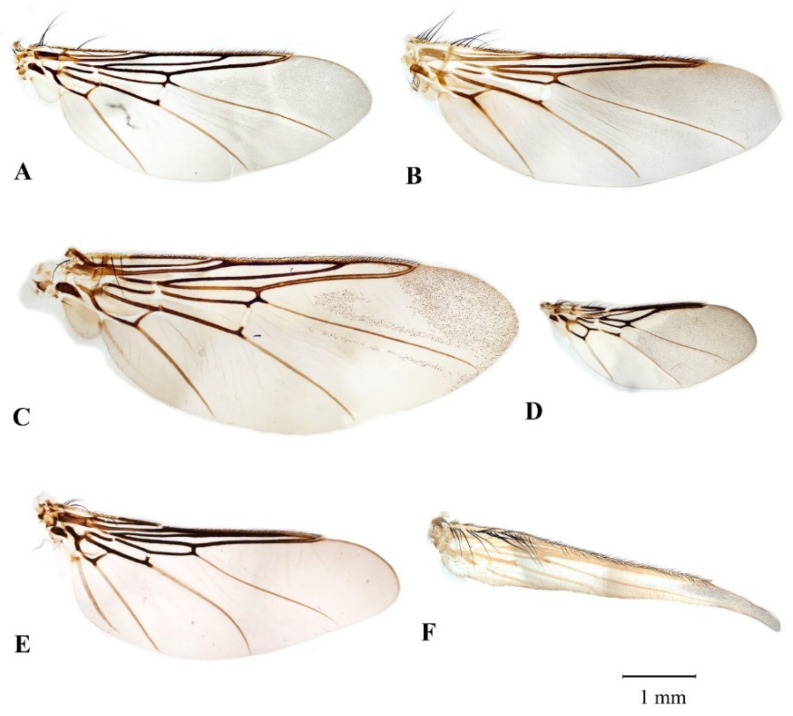
Wings of hippoboscid species caught during the study (photos by AS). Wing shape, length, and hair distribution are used for species determination: (**A**)—*Ornithomya fringillina*, (**B**)—*O. biloba*, (**C**)—*O. avicularia*, (**D**)—*Ornithoica turdi*, (**E**)—*Ornithophila metallica*, (**F**)—*Stenepteryx hirundinis*.

**Figure 2 microorganisms-10-00584-f002:**
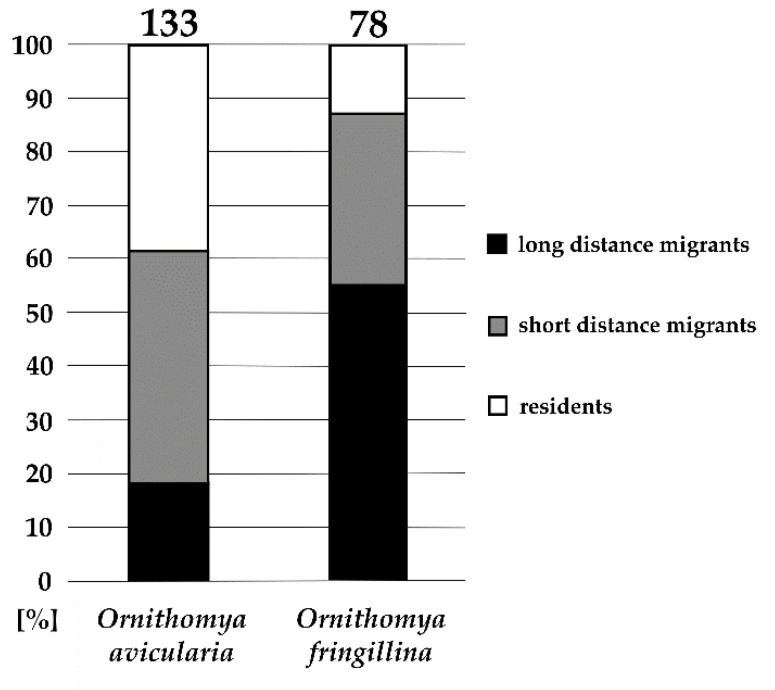
The occurrence of *Ornithomya* spp. on bird hosts in relation to host migratory status; numbers of collected specimens are given above columns.

**Figure 3 microorganisms-10-00584-f003:**
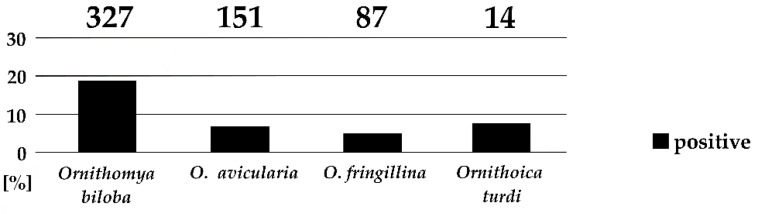
Prevalence of trypanosomes in *Ornithomya* spp. and *Ornithoica turdi*. Numbers of examined specimens are given above the columns.

**Figure 4 microorganisms-10-00584-f004:**
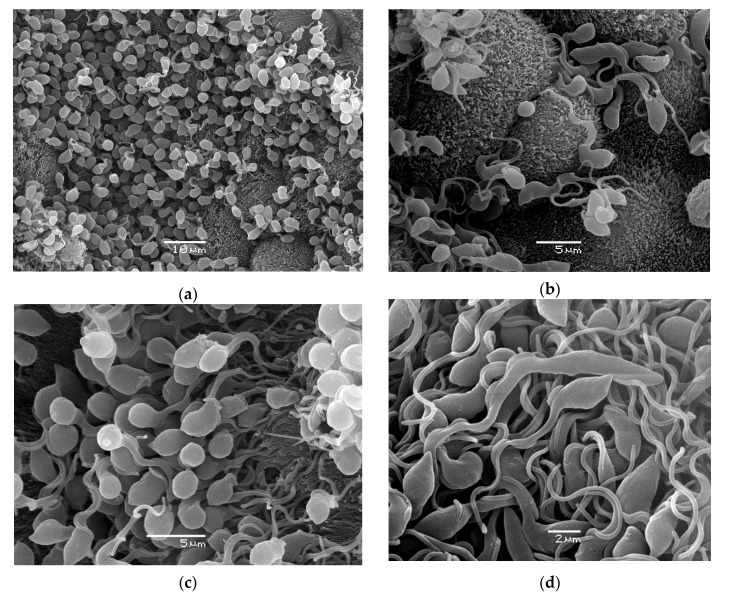
Trypanosomes in *O. avicularia* gut, natural infection. Cultivation of these resulted in a new strain OA23 identified as belonging to the new lineage B13. The female louse fly was caught on a juvenile thrush (*T. philomelos*) in Central Bohemia. Epimastigotes were attached to the gut epithelium (**a**–**c**) or were free, probably metacyclic forms (**d**).

**Figure 5 microorganisms-10-00584-f005:**
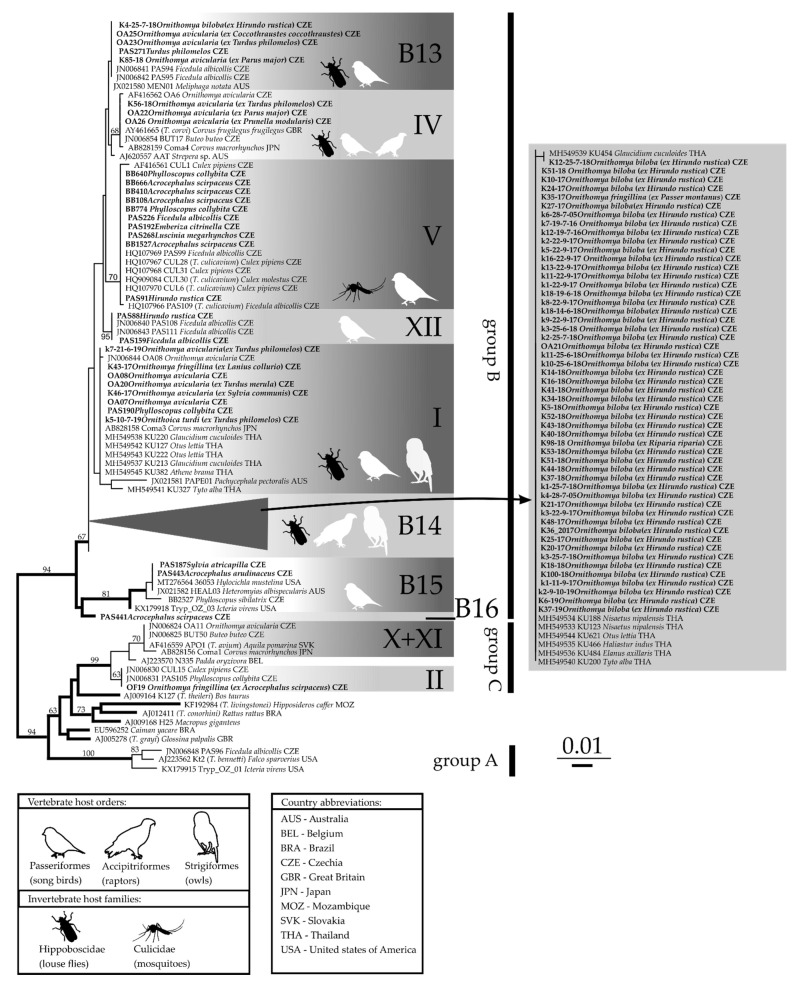
Maximum likelihood unrooted phylogenetic tree of avian trypanosome SSU rRNA gene sequences. The three groups of avian trypanosomes (A—*T. bennetti* group; B—*T. corvi/culicavium* group, and C—*T. avium/thomasbancrofti* group) along with previously described lineages (I–XII) are labelled after [16]; new lineages in the group B are labelled B13–B16. Original sequences are described in bold. The sequence description contains identifier, trypanosome host, and country of collection. Bootstrap support higher than 50 is shown at the branches.

**Table 1 microorganisms-10-00584-t001:** Avian hosts of hippoboscids according to the migratory status of the hosts (M): L—long-distance migrants, S—short-distance migrants, R—residents; Louse fly species: OA—*Ornithomya avicularia*, OF—*Ornithomya fringillina*, OB—*Ornithomya biloba*, OT—*Ornithoica turdi*, OM—*Ornithophila metallica*, SH—*Stenepteryx hirundinis*, CP—*Crataerina pallida*.

Avian Host	M	Louse Fly Species						
		OA	OF	OB	OT	OM	SH	CP
*Acrocephalus arundinaceus*	L	1						
*Acrocephalus palustris*	L	2	8					
*Acrocephalus schoenobaenus*	L	1						
*Acrocephalus scirpaceus*	L	4	21					
*Anthus trivialis*	L	1						
*Apus apus*	L							33
*Carduelis cannabina*	S		2					
*Coccothraustes coccothraustes*	S	5			4	1		
*Cyanistes caeruleus*	R	1	3	1				
*Delichon urbicum*	L						1	
*Dendrocopos major*	R	3						
*Dendrocopos medius*	R	6						
*Emberiza calandra*	S	1	2					
*Emberiza citrinella*	R	18	6		4			
*Emberiza schoeniclus*	S	1	1					
*Erithacus rubecula*	S		1					
*Ficedula albicollis*	L	4						
*Fringilla coelebs*	S	5						
*Hirundo rustica*	L	1	2	293			1	
*Jynx torquilla*	L	1	1					
*Lanius collurio*	L		2					
*Locustella luscinioides*	L	2						
*Luscinia svecica*	L	1						
*Motacilla alba*	S	2						
*Panurus biarmicus*	R	1						
*Parus major*	S	11	3					
*Passer montanus*	R	2	1	3				
*Phylloscopus collybita*	S	1	3					
*Picus viridis*	R	2						
*Prunella modularis*	S	7	3					
*Riparia riparia*	L	3		9				
*Sitta europaea*	R	3			1			
*Sturnus vulgaris*	S	2						
*Sylvia atricapilla*	S		10					
*Sylvia borin*	L	1	3					
*Sylvia communis*	L	2	3					
*Sylvia curruca*	L		3					
*Turdus merula*	R	15						
*Turdus philomelos*	S	23			5			
**Total**		**133**	**78**	**306**	**14**	**1**	**2**	**33**

## Data Availability

Sequences obtained in this study have been deposited in GenBank under the accession numbers OM509725-OM509732.

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
