# Peer review of "Avian Louse Flies and Their Trypanosomes: New Vectors, New Lineages and Host–Parasite Associations"

_microorganisms, 2022, doi:10.3390/microorganisms10030584_

Round 1
Reviewer 1 Report
Line 2. The authors could consider amending the title of the manuscript to better reflect their findings, for example: 'Novel avian trypanosome lineages and louse fly vectors identified in Czechia'.
Line 32. Delete '(Hippoboscidae)' from here, as it is already stated in Line 28.
Line 71. Replace 'team' with 'research group'.
Line 74. Replace 'zip-log' with 'zip-lock'.
Line 75. Replace 'in -20 C' with 'at -20 C'.
Line 98. Replace 'second step' with 'second round'
Line 177. Replace 'notorious' with 'known'.
Line 188. Replace '...from birds and vectors of...' to ' from birds; vectors of'
Line 294. Delete 'one'.
Figure 1. The authors could consider moving Figure 1 to the Methods section.
Figure 3. Only the positive bars should be shown in this figure.
Figure 5. The authors consider adding to the key the abbreviations of the countries, for example CZE Czechia, BRA Brazil, THA Thailand.
Reviewer 2 Report
The manuscript presents data on Hippoboscidae collected from birds in 3 different locations in the Czech Republic and the description of trypanosome lineages in these ectoparasites.
The work is sound and the data are valid and interesting and provide novel insight into the possible associations between hippoboscid flies, their hosts and the flagellated protozoa they harbor.
However, the manuscript needs to be improved in three points:
- The description of the methods must be clarified (the methods are valid; however, it took me until the Discussion to completely apprehend how the authors had come to their conclusions):
- The use of the expression “vector” is used too liberally. The authors have not made attempts to demonstrate transmission, so an insect harboring a protozoan cannot be considered a vector. The Discussion very nicely outlines the authors’ reason for considering vectorial capacity at least in some cases; however, this manuscript should not give the impression that the authors are not aware of this ongoing discussion in the scientific community, and refrain from using the term “vector” too freely.
- The language needs considerable improvement to comply with the standard of a scientific journal.
Therefore, major revision is recommended.
Below a list (following the order in the manuscript) of points that need the authors’ attention in a revision are listed.
- Abstract: l. 12: from reading the abstract it is not clear how the insects were collected (hippoboscids cannot be collected from the environment, but the collection during mist-netting of birds should be mentioned); l: 15 (and later, e.g. l. 151): The term “migrants” is usually used in a political context, “migratory bird species” is more appropriate. L. 18: as outlined above, the study design does not justify the term “vector” for all flies found. L.22: a definitive conclusion is missing in the abstract.
- Introduction: L. 30 should be changed to “to the parasitic lifestyle by… abdomen, claws for attachment to the host, and modified mouth parts”; l. 52: 5% is considered high by the authors, but in relation to what? L. 65f: Again, the finding of trypanosomes in the flies does not say much about their vectorial role and capacity. L. 66: the last sentence holds no information; it can be deleted.
- Materials and Methods: l. 74:”cold”: at what temperature (or temperature range)? L. 75 at -20°C, not in. All manufacturers must be named and the place and country where they are located must be given at their first mentioning, this must be added for e.g. Polysciences l. 78, TaKaRa l. 98, Thermo Scientific l. 104, and elsewhere. L. 82 and 89: the citations should be changed to “as described previously [xx]”. L. 88: which packages of R were used/which tests were applied?
- Results l. 126 and l. 141: Table, not Tab. The manuscript should also be reformatted in order not to leave half a page empty. The first paragraph on p. 4 can be moved up. L. 130: add full stop at the end of the figure caption. Is there a reference to a key available that can be cited here? L. 132: “were”, not “have been”.
- Table 1: what is missing (to me) in the Results and in the Discussion is the finding that almost 300 flies were found on a single species of birds (H. rustica), ands that the vast majority of the flies on this bird species belonged to one species, O. biloba, in contrast to much lower and usually more diverse flies on the other birds. Similarly, C. pallida was found only on A. apus and in much higher numbers compared to other fly species. Maybe the authors would like to comment on these findings. Does this have anything to do with the number of specimen caught for each bird species? This number may be included in the Table 1.
- 147: it would be beneficial to mention that only these two species were investigated for their preference of migratory birds because they were the most abundant species. It is suggested to change the sentence to “We evaluated differences in host preferences of the two most abundant louse fly species found, O. avicularis and O. fringillina, in terms of the migratory status of their bird hosts.” Fig. 2: The recurrent “%” can be deleted from the y-axis numbering, and added as a single description instead. Fig.2, caption: hosts, not host.
- 156: four of the seven collected louse fly species harboured…..l. 162: specific for swifts (Apus apus). l. 158ff and Fig. 3: Only the highest and lowest prevalence rates should be listed, and l. 159f “..while…species” can be deleted. Instead, Fig. 3 should show only the positive bars (since they add up to 100% with the negative ones, anyway), the total numbers and the exact percentages. The Figure caption must be corrected (…turdi. Numbers…).
- 166: “Trypanosome natural infections in…” first, the grammar is wrong, second, as outlined above, the presence of trypanosomes in the gut of louse flies after a blood meal is not equivalent to an infection. The authors reasons this in a very good manner in the discussion. However, they cannot use expression like “infection”, “vector” etc” beforehand without a proof that this is really the case. Alternative wordings must be used in all parts before this is resolved in the discussion.
- Figure 4: “natural infection” does not make sense here. Cultivation did not result in a strain, it revealed this strain after PCR analysis.
- Chapter “phylogenetic analysis”: Results must not contain references. The past tense must be used throughout. L. 182ff: The new lineages were found in louse flies, not in the bird hosts! Describing the parasites as from the birds is misleading, as no bird blood sampled were analysed. The authors must make sure that the results are described according to the methods of analysis. L. 187: same, “Two new trypanosome sequences isolated from birds??? There is no description of bird bold analysis anywhere.
- 5: The writing is so small that no details can be recognized. Which of the sequences were retrieved from this study, which were already deposited in Genbank? The authors should consider splitting it into an overview and then into the different groups or, in case of group B, into the different subgroups.
- Discussion: The point about the vectorial role is mentioned earlier and should be made clear at the beginning of the discussion (not in the middle, l. 260 ff.); the description of the electron microscopy results 8l. 265ff) should be moved to the Results (where so far only the microphotographs are located) to build a basis on the discussion on the vector role and infection vs. carrier). What is missing is a discussion of the role of the flies vs. the birds in the long-distance travel of the trypanosomes. The authors determined differences in host group preferences for two louse flies; however, is this really a key to the transmission of the trypanosomes, or are the birds the trypanosome reservoirs? L. 238ff: a parasite is specific for a host, not towards a host. L. 248: not area but distribution. L. 254: persists whether (not if). L. 257: more formal: which supports the hypothesis that the O. fringilline found in this study originated from northern breeding grounds”. l. 269: ranges are usually given from low to high numbers, not the other way round. L. 2268: “specific vectors” and l. 275 “opportunistic hosts”: the authors should be clear about their assumptions: What is a specific vector? One that transmits in acyclic manner, like glossinids transmit African tryps, as opposed to a mechanical transmission like tabanids transmit Tr. evansi? And why are they opportunistic hosts (and what type of vectors would they be then?)? l. 288: The authors should state clearly whether they assume that the B14 lineages were brought to Thailand by birds which host different louse flies in Europe and in Thailand but the same trypanosomes (which supports the idea that birds are the tryp reservoirs) or whether louse flies were brought to Thailand by birds and transmitted the tryps to birds that usually are hosts for other louse flies…. (which seems an odd scenario, but maybe the authors can comment of this).
- 298ff: The summary should clearly sum up all the conclusions from this work, and not repeat the results.
